# Observation of naturally canalized phonon polaritons in LiV$_2$O$_5$ thin layers

Ana I. F. Tresguerres-Mata [1,2,13], Christian Lanza [1,2,13], Javier Taboada-Gutiérrez [3], Joseph. R. Matson [4], Gonzalo Álvarez-Pérez [1,2,10], Masahiko Isobe [5], Aitana Tarazaga Martín-Luengo [1,2], Jiahua Duan [1,2,11,12], Stefan Partel [6], María Vélez [1,2], Javier Martín-Sánchez [1,2], Alexey Y. Nikitin [7,8], Joshua D. Caldwell [4,9] & Pablo Alonso-González [1,2] ✉

Polariton canalization is characterized by intrinsic collimation of energy flow along a single crystalline axis. This optical phenomenon has been experimentally demonstrated at the nanoscale by stacking and twisting van der Waals (vdW) layers of α-MoO$_3$, by combining α-MoO$_3$ and graphene, or by fabricating an h-BN metasurface. However, these material platforms have significant drawbacks, such as complex fabrication and high optical losses in the case of metasurfaces. Ideally, it would be possible to canalize polaritons "naturally" in a single pristine layer. Here, we theoretically predict and experimentally demonstrate naturally canalized phonon polaritons (PhPs) in a single thin layer of the vdW crystal LiV$_2$O$_5$. In addition to canalization, PhPs in LiV$_2$O$_5$ exhibit strong field confinement ($\lambda_p \sim \frac{\lambda_0}{27}$), slow group velocity (0.0015c), and ultra-low losses (lifetimes of 2 ps). Our findings are promising for the implementation of low-loss optical nanodevices where strongly directional light propagation is needed, such as waveguides or optical routers.

Canalization is an optical phenomenon characterized by intrinsic collimation of electromagnetic energy along a single crystal axis[1–3]. This phenomenon holds promise for the efficient guiding of light and therefore shows potential for a wide range of applications, such as directional thermal management, strong light–matter interactions, and integrated planar optics, among others. Canalization has been studied theoretically[4,5] and demonstrated experimentally[6–11] in the case of strongly confined polaritons[12–14], bringing this phenomenon to the nanoscale. Notably, this has been made possible by the emergence of different polaritonic media, including vdW materials such as h-BN[15–18], α-V$_2$O$_5$[19], and α-MoO$_3$[20,21], and other bulky crystals such as β-Ga$_2$O$_3$[22,23], or CdWO$_4$[24], all of them supporting highly anisotropic PhPs[25,26]–light coupled to lattice vibrations–. Specifically, canalization or extreme anisotropic propagation of PhPs has been observed in h-BN layers, allowing sub-diffraction imaging[27,28], theoretically predicted and experimentally demonstrated in heterostructures of graphene and

[1]Department of Physics, University of Oviedo, Oviedo 33006, Spain. [2]Center of Research on Nanomaterials and Nanotechnology, CINN (CSIC-Universidad de Oviedo), El Entrego 33940, Spain. [3]Department of Quantum Matter Physics, Université de Genève, 24 Quai Ernest Ansermet, CH-1211 Geneva, Switzerland. [4]Interdisciplinary Materials Science Program, Vanderbilt University, Nashville 37212 TN, USA. [5]Max-Planck Institute for Solid State Research, Stuttgart D-70569, Germany. [6]Vorarlberg University of Applied Sciences, Research Center of Microtechnology, Dornbirn, Austria. [7]Donostia International Physics Center (DIPC), Donostia/San Sebastián 20018, Spain. [8]IKERBASQUE, Basque Foundation for Science, Bilbao 48013, Spain. [9]Department of Mechanical Engineering, Vanderbilt University, Nashville 37235 TN, USA. [10]Present address: Center for Biomolecular Nanotechnologies, Istituto Italiano di Tecnologia, via Barsanti 14, Arnesano 73010, Italy. [11]Present address: Center for Quantum Physics, Key Laboratory of Advanced Optoelectronic Quantum Architecture and Measurement (MOE), School of Physics, Beijing, China. [12]Present address: Beijing Key Laboratory of Nanophotonics and Ultrafine Optoelectronic System, Beijing, Beijing Institute of Technology, Beijing, China. [13]These authors contributed equally: Ana I. F. Tresguerres-Mata, Christian Lanza. ✉ e-mail: pabloalonso@uniovi.es

$\alpha$-MoO$_3$[5,6], and in twisted $\alpha$-MoO$_3$ stacks[7–10], in which the excitation and coupling of in-plane hyperbolic PhPs has recently led to the discovery of broadband and all-angle tunable canalization[11]. However, despite these important achievements, these material platforms face critical drawbacks for the implementation of canalization in nanotechnological devices, including high optical losses in h-BN (inherent when considering out-of-plane extreme anisotropic propagation of PhPs, and extrinsic when considering in-plane canalization of PhPs in artificially engineered metasurfaces[29,30]), and the inherently complex fabrication of twisted stacks. An ideal solution to circumvent these drawbacks would imply the observation of polaritons that are intrinsically canalized and exhibit low losses in individual and pristine layers, which we coin as "naturally" canalized polaritons. However, although there are theoretical works predicting low-loss polariton canalization for the case of surface plasmons in individual metallic layers[31,32], naturally canalized polaritons remain unexplored experimentally. Moreover, they have not been addressed theoretically for the case of PhPs, or, more generally, for finite-thickness layers that can be considered 3D.

In this work, we theoretically predict and provide experimental evidence of naturally canalized PhPs in an individual thin layer of the vdW crystal LiV$_2$O$_5$. Interestingly, the origin of such natural canalization is found to be rooted in the strongly anisotropic in-plane permittivity of LiV$_2$O$_5$, which is extracted by accurate modeling of far-field spectroscopy data. Importantly, in addition to canalization, PhPs in LiV$_2$O$_5$ exhibit strong field confinement ($\lambda_p \sim \frac{\lambda_0}{27}$) and ultra-low losses (lifetimes up to 2 ps).

## Results and discussion

### Theory of naturally canalized PhPs in thin dielectric layers

Firstly, to theoretically study the phenomenon of natural canalization of PhPs, we conduct numerical simulations of PhP propagation on a thin dielectric layer along the two in-plane principal directions $x$ and $y$. We consider three cases characterized by representative diagonal permittivity tensors. In the first case (isotropic), we select $\varepsilon_x = \varepsilon_y = 5 + 0.05i$, and $\varepsilon_z = -5 + 0.05i$, i.e., a diagonal tensor in which its principal components have the same moduli (note, however, that at least one of them must be negative to fulfill the conditions needed to support polaritons. The choice of two positive principal components is intended to highlight the main dielectric behavior of the layer). By plotting the norm of the simulated electric field $|E|$ (Fig. 1a), we observe PhPs propagating along all in-plane directions with the same wavelength (see inset for a plot of the real part, $Re(E_z)$). Such isotropic propagation can be better visualized by plotting the isofrequency curve (IFC) for the in-plane momentum ($q_x = k_x/k_0$, $q_y = k_y/k_0$, with $k_0$ the wavevector of the incident field) extracted by taking the Fourier transform (FT) of the real space image (Fig. 1d). The resulting IFC represents a circle of radius $\sim |q| = 20$, which illustrates the isotropic propagation of strongly confined PhPs with the same wavevector, $\mathbf{q}$, that is collinear to the energy flux $\mathbf{S}$ ($\mathbf{S}$ results from taking the normal to the IFC, indicated by white arrows in the figure) along all in-plane directions. In the second case (anisotropic), we study the PhP propagation when $\varepsilon_x = 5 + 5i$, $\varepsilon_y = 5 + 0.05i$, and $\varepsilon_z = -5 + 0.05i$, i.e., when one of the diagonal components of the permittivity tensor presents a much larger imaginary part ($Im(\varepsilon_x) \gg Im(\varepsilon_{y,z})$) implying a much larger absorption along one

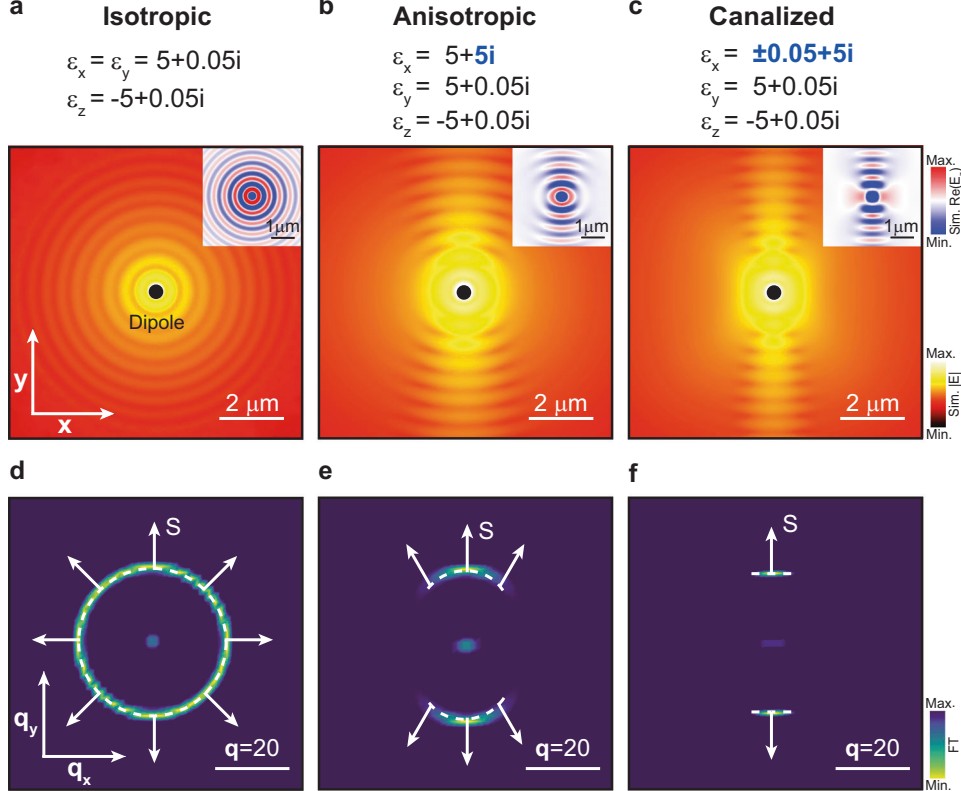

**Fig. 1 | Naturally canalized phonon polaritons (PhPs) in a thin dielectric layer.** **a** Norm of the electromagnetic field $|E|$ numerically calculated in a layer (thickness $d = 200$ nm) with permittivity $\varepsilon_x = \varepsilon_y = 5 + 0.05i$, and $\varepsilon_z = -5 + 0.05i$. A vertically oriented point dipole is employed as polaritonic excitation source. Clear isotropic propagation of strongly confined PhPs is observed. Inset: real part of the out-of-plane component ($E_z$) of the electromagnetic field. **b** Same as **a** for a permittivity $\varepsilon_x = 5 + 5i$, $\varepsilon_y = 5 + 0.05i$, and $\varepsilon_z = -5 + 0.05i$. Anisotropic (arc-shaped) polariton propagation centered along the y-axis is observed. **c** Same as **a** and **b** for a permittivity $\varepsilon_x = 0.05 + 5i$, $\varepsilon_y = 5 + 0.05i$, and $\varepsilon_z = -5 + 0.05i$. Naturally canalized PhPs propagate along the y-axis. **d–f** Isofrequency curve (IFC) obtained by Fourier transforming (FT) the real-space simulation in (**a–c**), respectively. The white arrows indicate the energy flux $\mathbf{S}$.

direction of the layer. By plotting $|E|$ (Fig. 1b), we observe the propagation of strongly confined PhPs along an angular sector of in-plane directions close to the $y$-axis (see inset for a plot of the real part, $Re(E_z)$). The corresponding IFC (Fig. 1e) consists of an arc-shaped curve, indeed showing that only certain in-plane directions close to the $y$-axis exhibit available optical states. With the aim of enhancing such directional propagation of PhPs, we study in the last case (canalized) the PhP propagation when $\varepsilon_x = \pm 0.05 + 5i$, $\varepsilon_y = 5 + 0.05i$, and $\varepsilon_z = -5 + 0.05i$, i.e., when one of the diagonal components, $\varepsilon_x$, not only exhibits a larger imaginary part but also a relatively smaller real part (i.e., the mode is overdamped), such that $Re(\varepsilon_x) \ll Im(\varepsilon_x)$. The $\pm$ sign in $\varepsilon_x$ accounts for the fact that the sign does not play an appreciable role in the simulation, as long as it is close to zero and the imaginary part dominates. By plotting $|E|$ for these permittivity values (Fig. 1c), we observe the excitation of strongly confined PhPs propagating mainly along one direction, the $y$-axis (when we mention along one direction, PhPs propagate in two senses along that same axis), i.e., along the perpendicular direction to that where the permittivity has drastically changed (see inset for a plot of the real part, $Re(E_z)$). This strongly directional propagation is clearly observed by plotting the corresponding IFC (Fig. 1f), showing two almost parallel IFC contours separated by a distance $\sim |q| = 40$. This peculiar geometry, in which for all available wavevectors, there is a unique direction of the energy flux $\mathbf{S}$ (white arrows), has been previously observed for PhPs in twisted vdW stacks and constitutes a clear signature of polariton canalization. Our modeling, therefore, predicts that the combination of a large imaginary part of the permittivity (large optical absorption) and a real part tending to zero (reduced field penetration) along a given in-plane direction in a thin dielectric layer can induce the natural canalization of PhPs along the perpendicular axis.

Note that although material absorption canalization has been studied for 2D plasmonic layers[31,32], the case detailed above involves the consideration of a thin slab with finite thickness ($d = 200$ nm), which optically means that the out-of-plane component of the permittivity tensor must be considered analytically (see "Methods"). Our results therefore demonstrate the possibility of translating this canalization concept to a 3D system supporting PhPs.

## Crystal and optical properties of LiV$_2$O$_5$

In the following, we introduce the vdW crystal LiV$_2$O$_5$ (Fig. 2) by describing its crystal structure and extracting its optical permittivity. Resulting from the intercalation of α-V$_2$O$_5$ with the alkaline atom Li[33,34] (see "Methods"), LiV$_2$O$_5$ has an orthorhombic crystal lattice consisting of VO$_5$ pyramids that are connected along their edges forming zigzag chains along the $b$ direction of the crystal (Fig. 2a). The lattice parameters are $a = 0.9702$ nm, $b = 0.3607$ nm, and $c = 1.0664$ nm along the [100], [001], and [010] crystal directions, respectively[34], thus revealing a strongly anisotropic crystal structure. Although both LiV$_2$O$_5$ and α-V$_2$O$_5$ have orthorhombic crystal unit cells, LiV$_2$O$_5$ has a *Pnma* space group, whereas the space group of α-V$_2$O$_5$ is *Pmmn*. To study the optical properties of LiV$_2$O$_5$, we carry out Fourier-Transform Infrared Spectroscopy (FTIR) reflectance measurements with the incident light polarized along the $a$ and $b$ crystal directions (blue and green curves in Fig. 2b, respectively, see Supplementary Figs. 1 and 2). The inset in Fig. 2b shows an optical image of the LiV$_2$O$_5$ layer used for these measurements and placed on a BaF$_2$ substrate, which is transparent in the spectral range of our measurements[35]. Previous studies provided the initial dielectric function fit values[34]. To ensure robust agreement, these values were then tailored to align with our experimental spectra. By fitting the resulting spectra with a Lorentz model of coupled oscillators[35] (black dashed curves in Fig. 2b), we derived the dielectric permittivity tensor along the three crystallographic directions. In particular, we extract the parameters $\omega_{TO}$, $\omega_{LO}$, $\gamma_{TO}$, and $\gamma_{LO}$ (Fig. 2c, d, see Supplementary Table 1 and

Supplementary Fig. 3), i.e., the frequencies of the transverse (TO) and longitudinal optical (LO) phonons together with the corresponding damping rate. Interestingly, by plotting the real and imaginary part of the extracted permittivity along the $a$, $b$, and $c$ crystal directions (blue, green, and red curves in Fig. 2c, d, respectively), we observe several spectral bands in which at least one of the permittivity components is negative and thus PhPs can be excited. Specifically, we distinguish three spectral bands (typically called Reststrahlen bands (RBs) that are bound between the TO and LO frequencies): RB$_{a1}$ between $\omega \sim 948$ cm$^{-1}$ and $\omega \sim 964$ cm$^{-1}$ and RB$_{a2}$ between $\omega \sim 1005$ cm$^{-1}$ and $\omega \sim 1012$ cm$^{-1}$ for the $a$ direction, and RB$_c$ between $\omega \sim 980$ cm$^{-1}$ and $\omega \sim 1024$ cm$^{-1}$ for the $c$ direction. Importantly, we observe that within RB$_{a2}$ (black squared range in Fig. 2c, d) the permittivity along the $a$ direction has its real part close to zero while the imaginary part is two orders of magnitude larger than along the perpendicular $b$ direction, i.e., $Re(\varepsilon_a) \ll Im(\varepsilon_a)$ and $Im(\varepsilon_a) \gg Im(\varepsilon_b)$. This combination of permittivity values fulfills the criterion extracted in Fig. 1 for the observation of naturally canalized PhPs in a thin dielectric layer. Here, the negative out-of-plane permittivity induces propagation along the directions in the plane that show positive permittivity. This is in stark contrast to prior reports on strongly in-plane anisotropic (hyperbolic) PhPs in vdW materials, such as α-V$_2$O$_5$[19] and α-MoO$_3$[20,21], where propagation takes place along the in-plane directions that feature negative values, due to the positive sign of the out-of-plane permittivity.

## Analytical calculations of PhP propagation in LiV$_2$O$_5$

To better analyze the potential natural canalization of PhPs in LiV$_2$O$_5$, we next perform an analytical study (Fig. 3) using the explicit expression for the dispersion of highly confined polaritons in anisotropic layers[36] (see "Methods" and Supplementary Fig. 4), in which we introduce the dielectric permittivity tensor extracted in Fig. 2. We plot analytical IFCs (Fig. 3a, b) for PhPs excited in a LiV$_2$O$_5$ thin layer of thickness $d = 200$ nm at a frequency range between 995 cm$^{-1}$ and 1010 cm$^{-1}$, i.e., in a frequency range that partially covers RB$_c$ and completely RB$_{a2}$, where canalization is expected according to our previous analysis. We observe closed curves formed by wavevectors much larger than $k_0$, indicating the sub-diffractional nature of PhPs in LiV$_2$O$_5$. Furthermore, we observe that while the IFC for the real part of $q_{a/b} = k_{a/b}/k_0$ ($Re(q)$, Fig. 3a) evolves from elliptic to rectangular with increasing frequency, the IFC for the imaginary part of $q_{a/b} = k_{a/b}/k_0$ ($Im(q)$, Fig. 3b) undergoes a transition from elliptic to peanut-shaped. This result can be better visualized by highlighting two characteristic 2D cross-sections in the IFCs at frequencies $\omega = 997$ cm$^{-1}$ and $\omega = 1007$ cm$^{-1}$ (black and red curves for $Re(q)$ and $Im(q)$, respectively). More importantly, by plotting together the curves for $Re(q)$ and $Im(q)$ at these frequencies (Fig. 3c, d), we observe that while at $\omega = 997$ cm$^{-1}$, the elliptic curve for $Re(q)$ is always significantly larger than the elliptic curve for $Im(q)$, at $\omega = 1007$ cm$^{-1}$, the rectangular curve for $Re(q)$ is significantly larger than the peanut-shaped curve for $Im(q)$ only for a small angular domain of directions close to the $b$ direction. Since the condition $Re(q) > Im(q)$ can serve as an indication of the propagative nature of polaritonic modes, these calculations predict a polariton canalization at $\omega = 1007$ cm$^{-1}$. To corroborate these results, we numerically calculate the IFCs at the same frequencies (color plots in Fig. 3c, d, see "Methods" for details). At $\omega = 997$ cm$^{-1}$ we obtain an elliptic IFC, indicating the propagation of PhPs along all in-plane directions with slightly different wavelengths. On the other hand, at $\omega = 1007$ cm$^{-1}$ we obtain a very different IFC consisting of two linear segments parallel to the $a$ direction of LiV$_2$O$_5$. As discussed in relation to Fig. 1f, this type of IFC constitutes a clear fingerprint of canalization, which in this

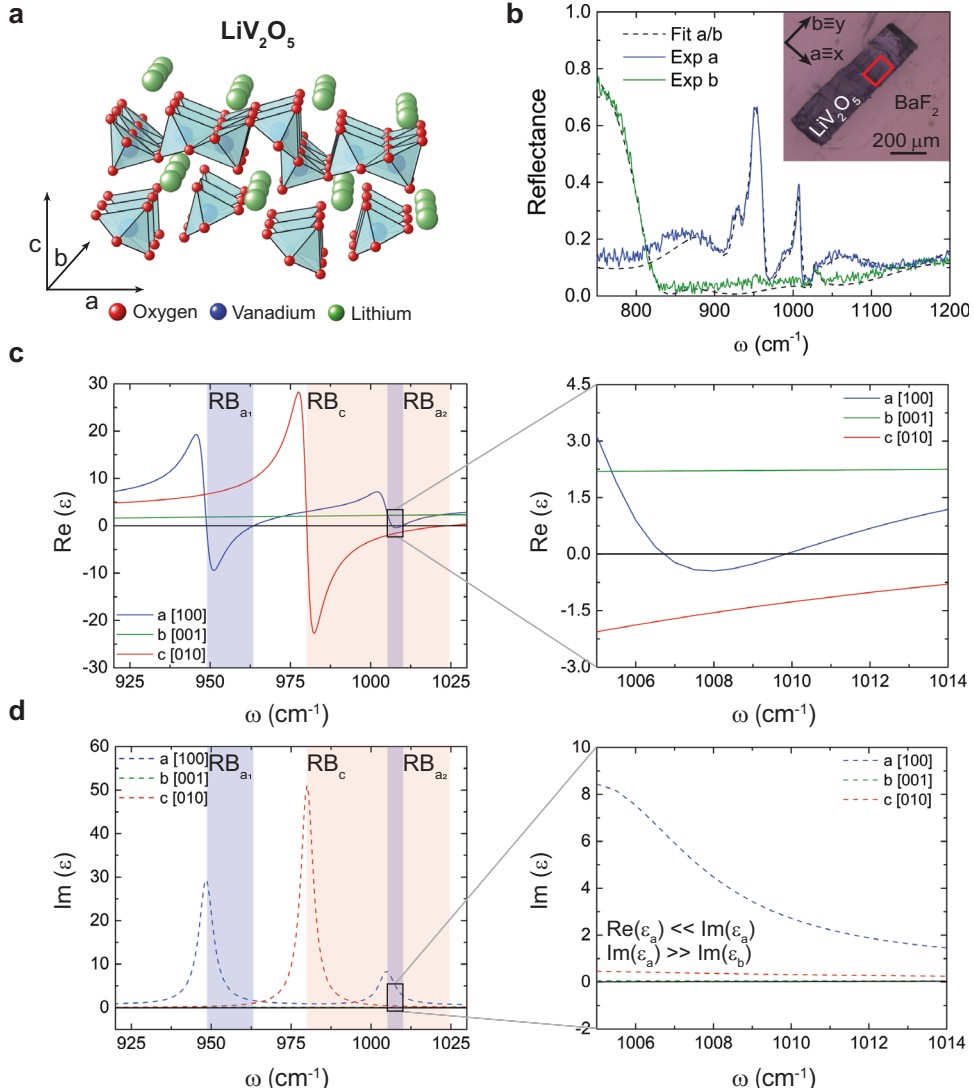

**Fig. 2 | Crystal structure and optical properties of LiV$_2$O$_5$. a** Illustration of the orthorhombic lattice structure of LiV$_2$O$_5$ (red spheres: oxygen atoms, blue spheres: vanadium atoms, green spheres: lithium atoms). The crystal structure is formed by distorted VO$_5$ pyramids connected along their edges, forming zigzag chains along the $b$ = [001] direction. **b** Reflectance spectra from Fourier-Transform Infra-red Spectroscopy (FTIR) along the $a$ = [100] and $b$ directions (blue, and green curves, respectively) as indicated in the optical image of the inset (the red square indicates the measurement area). Fittings of the spectra using a Drude–Lorentz model are indicated by black dashed lines. The thickness of the LiV$_2$O$_5$ layer is $d$ = 6.43 μm, and the substrate is BaF$_2$. **c, d** Real (solid curves) and imaginary (dashed curves) components $\varepsilon_{a/b/c}$ of the permittivity along the $a$ (blue), $b$ (green), and $c$ = [010] (red) crystal directions, extracted from the fittings in (**b**). The shadowed regions represent the three Reststrahlen bands (RBs) at this spectral range, including RB$_{a1}$ and RB$_{a2}$ for the $a$ direction (blue) and RB$_c$ for the $c$ direction (red). The black squares indicate a zoom-in area in which the permittivity function (right panels) fulfills the conditions extracted in Fig. 1 for the observation of naturally canalized PhPs, namely $Re(\varepsilon_a) \ll Im(\varepsilon_a)$ and $Im(\varepsilon_a) \gg Im(\varepsilon_b)$.

case occurs along the $b$ direction of LiV$_2$O$_5$. However, while the FT-calculated color plot shows an open contour owing to polariton damping by intrinsic material losses, we note here that the analytical IFC does show a closed shape across the whole frequency range plotted in Fig. 3a–d. This evidences that polariton canalization is induced here by the epsilon-near-zero behavior along the $a$ direction together with the high damping by intrinsic material losses, also along this direction. This therefore provides a fundamental difference to prior demonstrations of canalization in vdW materials, such as in heterostructures of graphene and α-MoO$_3$[5,6], and in twisted α-MoO$_3$ stacks[7–11], where canalization stems from a topological transition from open to closed IFCs. As such, our work provides an innovative mechanism for polariton canalization that is not rooted in seeking a topological transition of the IFC, but a combination of large imaginary part and close-to-zero real part of the permittivity.

To further analyze this canalization phenomenon, we plot the ratio $\left|\frac{Re(q)}{Im(q)}\right|$ (considered as a valid figure-of-merit (FOM) or quality factor of the propagation of polaritons) along the $a$ and $b$ directions within the spectral range from 995 cm$^{-1}$ to 1020 cm$^{-1}$ (blue and green curves in Fig. 3e). Interestingly, we observe that in the spectral band corresponding to RB$_{a2}$, the FOM reaches values of up to ~ 5 along the $b$ direction, while it takes values close to 1 along the $a$ direction. This result confirms a preferential propagation along the $b$ direction within RB$_{a2}$, which can be better visualized by performing numerical simulations at $\omega$ = 1007 cm$^{-1}$ (inset inside RB$_{a2}$ in the figure, see "Methods" for details). On the contrary, an analogous plot calculated at a frequency outside RB$_{a2}$, e.g., at $\omega$ = 997 cm$^{-1}$, shows elliptic propagation of PhPs along all in-plane directions (inset outside RB$_{a2}$ in the figure). To get further insights into the canalization phenomenon and its

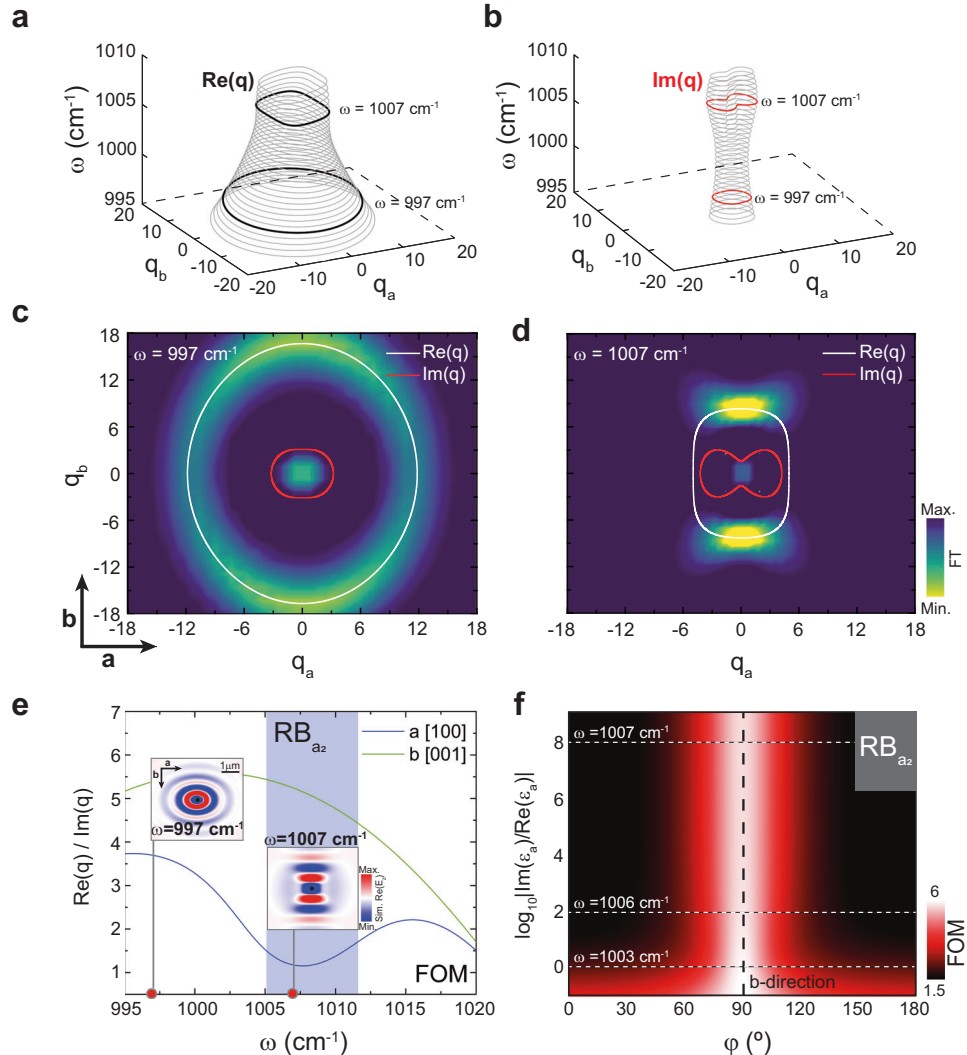

**Fig. 3 | Study of the propagation of PhPs in LiV$_2$O$_5$. a** Analytical 3D IFC ($q_a$,$q_b$,$\omega$) for the real part of the polariton wavevector ($Re(q)$) at frequencies $\omega = 997$ cm$^{-1}$ and $\omega = 1007$ cm$^{-1}$. **b** Analytical 3D IFC ($q_a$,$q_b$,$\omega$) for the imaginary part of the polariton wavevector ($Im(q)$) at frequencies $\omega = 997$ cm$^{-1}$ and $\omega = 1007$ cm$^{-1}$. **c**, Analytical 2D IFC extracted from combining $Re(q)$ (white) and $Im(q)$ (red) from (a and b) at $\omega = 997$ cm$^{-1}$. $Re(q)$ is larger than $Im(q)$ along all directions. The color plot is obtained by Fourier Transforming (FT) a numerical simulation in real space of PhPs excited using a point dipole on a LiV$_2$O$_5$ layer with thickness $d = 200$ nm at $\omega = 997$ cm$^{-1}$. **d** Analytical 2D IFC extracted from combining $Re(q)$ (white) and $Im(q)$ (red) from (a and b) at $\omega = 1007$ cm$^{-1}$. $Re(q)$ is larger than $Im(q)$ only in a significant manner along the $b$ direction, predicting canalization of PhPs along this axis. The color plot is obtained by Fourier Transforming (FT) a numerical simulation in real space of PhPs excited using a point dipole on a LiV$_2$O$_5$ layer with thickness $d =$ 200 nm at $\omega = 1007$ cm$^{-1}$. The appearance of two flat bands at $\omega = 1007$ cm$^{-1}$ corroborates the existence of naturally canalized PhPs in LiV$_2$O$_5$. **e** Figure of Merit (FOM) for propagating PhPs, defined as $Re(q)/Im(q)$, along the $a$ (blue) and $b$ (green) crystal directions. The approximately 5× difference in the FOM between the $b$ and $a$ directions corroborates the canalization of PhPs along the former. The small insets show the real part of the electromagnetic field, $Re(E_z)$, from numerical simulations, showing canalized PhPs at $\omega = 1007$ cm$^{-1}$. The blue shaded area indicates the Reststrahlen band $a_2$ (RB$_{a2}$). **f,** PhPs quality factor $\left|\frac{Re(q)}{Im(q)}\right|$ as a function of in-plane propagation angle $\varphi$ (°) and the permittivity ratio along the $a$ direction $\left|\frac{Im(\varepsilon_a)}{Re(\varepsilon_a)}\right|$. The white dashed lines correspond to the permittivity ratio value at $\omega = 1003$ cm$^{-1}$, $\omega = 1006$ cm$^{-1}$, and $\omega = 1007$ cm$^{-1}$, while the black dashed line traces the $b$ direction.

intrinsic relation to the combination of in-plane components of the crystal permittivity, we can also calculate the FOM as a function of in-plane angle and frequency-dependent material absorption along the $a$ direction, defined as $\left|\frac{Im(\varepsilon_a)}{Re(\varepsilon_a)}\right|$ (Fig. 3f). For the sake of clarity, the normalized absorption is represented on a logarithmic scale. We observe that if absorption increases along the $a$ direction (coinciding with the spectral range covered by RB$_{a2}$) the polariton propagation is much more restricted to directions closer to $\varphi = 90°$ or, equivalently, to the crystal $b$ direction (bright red color). Altogether, our theoretical results confirm the possibility of observing naturally canalized PhPs in

LiV$_2$O$_5$, which stems from the high damping of this material along the $a$ direction, together with its epsilon-near-zero behavior, thereby introducing an original mechanism for polariton canalization in vdW materials.

## Nanoimaging of naturally canalized PhPs in LiV$_2$O$_5$

To experimentally study the excitation and propagation of PhPs in LiV$_2$O$_5$, we carried out near-field s-SNOM (scattering-type scanning near-field optical microscopy) polariton imaging (see "Methods" and Supplementary Figs. 5 and 6). This involved examining two thin layers of LiV$_2$O$_5$, one with a thickness of 213 nm and the other with a smaller thickness of 116 nm, at incident frequencies

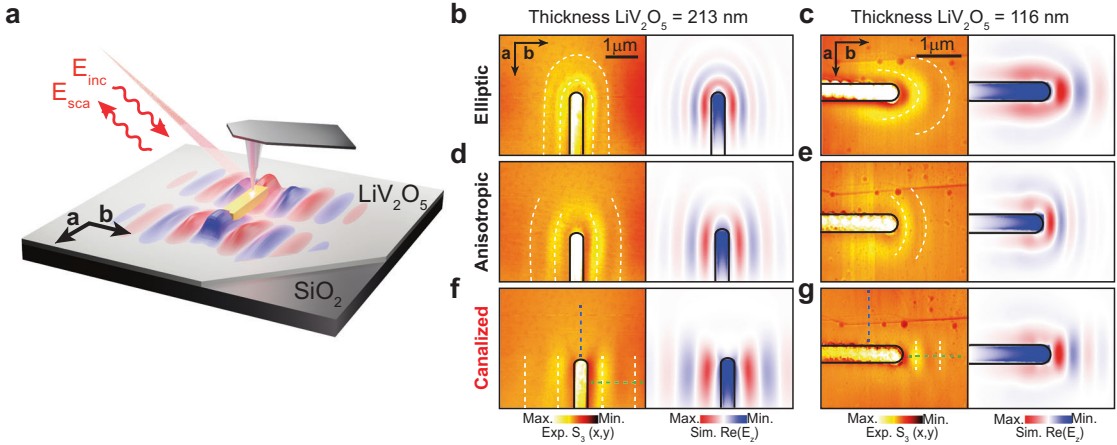

**Fig. 4 | Observation of naturally canalized PhPs in thin LiV$_2$O$_5$ layers.**
**a** Schematics of the near-field visualization of PhPs in LiV$_2$O$_5$ placed on a SiO$_2$ substrate. $E_{inc}$ and $E_{sca}$ represent the incident and scattered electromagnetic field. A resonant Au-antenna illuminated with s-polarized infrared light efficiently launches PhPs, which are probed by an s-SNOM (scattering-type scanning near-field optical microscopy) microscope employing a metallic tip. **b**, **c** Near-field amplitude s-SNOM images ($s_3$) and numerically simulated images ($Re(E_z)$) for LiV$_2$O$_5$ thin layers with thicknesses of 213 nm and 116 nm, respectively. The excited PhPs exhibit elliptic propagation at $\omega = 997$ cm$^{-1}$ and $\omega = 1010$ cm$^{-1}$, respectively. **d**, **e** Same as **b**, **c** for $\omega = 1003$ cm$^{-1}$. The excited PhPs exhibit anisotropic propagation. **f**, **g** Same as **b**, **c** for $\omega = 1007$ cm$^{-1}$. The excited PhPs exhibit canalized propagation. White dashed lines are guides to the eye in all panels.

covering the spectral bands RB$_c$ and RB$_{a2}$. For the excitation of PhPs, we fabricate resonant Au nanoantennas on the LiV$_2$O$_5$ layers, which, when they are illuminated with s-polarization, act as very efficient polaritonic sources (see schematics in Fig. 4a). Importantly, this optical scheme allows the s-SNOM tip to act primarily as a polaritonic probe[28], enabling direct real-space visualization of the propagation of PhPs excited by the nanoantenna (note that typical s-SNOM polariton imaging is carried out under optical schemes in which the tip plays a critical role for both launching and detection, resulting in interferometric patterns). Figure 4b–g show the resulting images at illuminating frequencies within RB$_c$ ($\omega = 997$ cm$^{-1}$ and $\omega = 1003$ cm$^{-1}$, Fig. 4b, d for the 213 nm flake; $\omega = 1010$ cm$^{-1}$ and $\omega = 1003$ cm$^{-1}$, Fig. 4c, e respectively for the 116 nm flake) and RB$_{a2}$ ($\omega = 1007$ cm$^{-1}$, Fig. 4f, g for both samples) for two LiV$_2$O$_5$ flakes of different thicknesses. The thinner flake is inspected with the Au-antenna rotated 90°. For both RBs, we observe polaritonic fringes with small wavelengths in the range of 1 μm (~$\lambda_0$/16 for RB$_c$ and ~$\lambda_0$/8 for RB$_{a2}$ for the 213 nm flake), revealing the sub-diffraction nature of PhPs in LiV$_2$O$_5$. However, we observe a very different propagation along the two in-plane directions for the three frequencies. At $\omega = 997$ cm$^{-1}$, the near-field amplitude image reveals PhPs emanating from the Au nanoantenna with elliptic wavefronts, exhibiting their smallest wavelength (~0.6 μm) along the $b$ direction and their largest (~0.8 μm) along the orthogonal $a$ direction for the thicker flake (Fig. 4b). At $\omega = 1003$ cm$^{-1}$, the propagation of PhPs remains elliptic but becomes highly anisotropic (Fig. 4d). More strikingly, at $\omega = 1007$ cm$^{-1}$, we observe that the propagation of PhPs occurs with a fixed wavelength ($\sim 1.2$ μm and $\sim 0.5$ μm for the thicker and thinner flake, respectively) along one single axis corresponding to the $b$ direction in LiV$_2$O$_5$ (Fig. 4f, g). For the thinner flake, we observe the same evolution towards canalization in the $b$ direction (Fig. 4c, e, g). A different PhP wavelength is, however, observed, as expected due to the effect of a different thickness in the PhPs dispersion. To theoretically validate these results, we conducted numerical simulations to mimic the experiment. The resulting plots ($Re(E_z)$), displayed next to the near-field amplitude images for all cases, exhibit a good agreement with the experiment. This allows us to confirm the excitation of elliptic, anisotropic, and, more importantly, naturally canalized PhPs in LiV$_2$O$_5$.

## Dispersions and lifetimes of PhPs in thin layers of LiV$_2$O$_5$
For a better understanding and quantitative analysis of the propagation of PhPs in LiV$_2$O$_5$, we extracted the PhP dispersions, $\omega(q_i)$ ($i = a, b$) from monochromatic s-SNOM images (not shown) of the layer with a thickness of 213 nm shown in Fig. 4b, d, f. The dispersions, extracted within the spectral range from 993 cm$^{-1}$ to 1011 cm$^{-1}$, are plotted in Fig. 5a, b. In both plots we corroborate the sub-diffractional nature of the PhPs, particularly along the $b$ direction (Fig. 5b), where the wavevector takes values up to $1.7 \times 10^5$ cm$^{-1}$, meaning a wavelength confinement of $\sim \lambda_0/27$. Furthermore, we observe very different tendencies for the dispersions along the two in-plane directions: while for the $b$ direction, the PhP wavevector always decreases with increasing $\omega$ (corresponding to a negative phase velocity[20]), this dependence only occurs below 1005 cm$^{-1}$ for the $a$ direction (Fig. 5a). More importantly, we observe the formation of a spectral gap along the $a$ direction in which no propagation of PhPs is visualized, which is corroborated by performing transfer-matrix (TM) calculations[37] (color plot, see "Methods"). Interestingly, this gap coincides with RB$_{a2}$, as revealed by indicating $\omega_{TO}$ and $\omega_{LO}$ (solid white lines) extracted for RB$_{a2}$ in Fig. 2. This observation is in agreement with our analysis in Fig. 3, and particularly with the dependence of the polariton quality factor with the permittivity ratio $\left| \frac{Im(\varepsilon_a)}{Re(\varepsilon_a)} \right|$, allowing us to confirm the high absorption along the $a$ direction in RB$_{a2}$ as the origin of canalization of PhPs along the perpendicular $b$ direction.

Finally, we calculate the lifetimes of PhPs in LiV$_2$O$_5$, a key property for their potential implementation in sensing applications or to enhance non-linear processes at the nanoscale. To do this, we take a line profile along the $b$ (Fig. 5c) and $a$ (Fig. 5d) directions in the s-SNOM images taken at $\omega = 1007$ cm$^{-1}$ (green and blue dashed lines, respectively, in Fig. 4f, g) and perform a fitting to a damped sinewave (red solid lines; more information is given in Supplementary Figs. 7,8). By extracting from these fittings the PhPs propagation lengths, $L_p \sim 1$ μm (thicker flake) and $L_p \sim 0.5$ μm (thinner flake), and using the group velocities, $v_g = 0.0015c$ (thicker flake) and $v_g = 0.0007c$ (thinner flake), extracted by fitting the dispersions, we obtain a similar lifetime $\tau = L_p/v_g = 2 \pm 0.5$ ps for both flakes. Such a long lifetime, similar to the lifetimes measured for PhPs in isotopically enriched h-BN[17] and of the same order as the lifetime recently reported for PhPs in α-MoO$_3$[20]

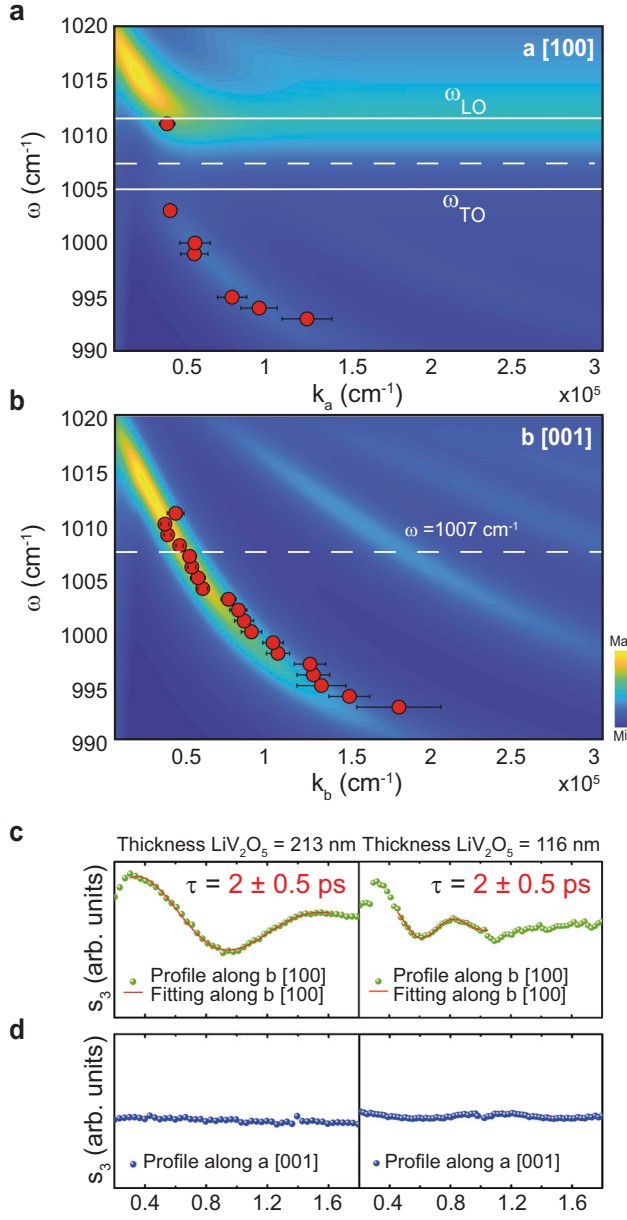

**Fig. 5 | PhPs dispersion and lifetimes in thin LiV$_2$O$_5$ layers. a** Experimental (red dots) PhP dispersion extracted by s-SNOM imaging along the *a* direction of the LiV$_2$O$_5$ layer (213 nm) measured in Fig. 4. Experimental points are shown with error bars that take into account the experimental and fitting errors. Theoretically calculated dispersion (using transfer matrix (TM) formalism) is shown as a color plot. A good agreement with the experiment is obtained. The white horizontal solid lines indicate $\omega_{TO}$ and $\omega_{LO}$ defining RB$_{a2}$, where canalized PhPs are excited. **b** Same as **a** for the *b* direction in LiV$_2$O$_5$. A good agreement with the experiment is obtained. The white dashed line indicates the frequency of canalization of PhPs. **c, d** s-SNOM line traces along the *b* and *a* directions (Fig. 5c, d, respectively) of the images shown in Fig. 4f, g at the canalization frequency $\omega$ = 1007 cm$^{-1}$ on LiV$_2$O$_5$ thin layers with thicknesses 213 nm and 116 nm. Fitting curves of the line traces along the *b* direction using a damped sine wave function are shown as red solid lines to extract the lifetimes ($\tau$).

and α-V$_2$O$_5$[19], reveals the low loss nature of canalized PhPs in LiV$_2$O$_5$. We note that, like PhPs in α-MoO$_3$ and α-V$_2$O$_5$, these long lifetimes do not imply long propagation lengths and thus the visualization of numerous polaritonic oscillations in our s-SNOM experiments. As previously reported[19,20], this effect is rather

common in highly confined PhPs in thin slabs and results from their very slow group velocity.

In conclusion, our work introduces theoretically and demonstrates experimentally a type of polaritonic mode: naturally canalized PhPs, which stem from an original mechanism for polariton canalization in vdW materials. Observed by introducing the vdW crystal LiV$_2$O$_5$ as a polaritonic medium, which we accurately characterize here, naturally canalized PhPs in LiV$_2$O$_5$ stem from a combination of high damping and epsilon-near-zero behavior along the *a* crystal direction. In addition to canalization, PhPs in LiV$_2$O$_5$ also features strong confinement (up to ~$\lambda_0$/27), slow group velocities (0.0015c), and long lifetimes (2 ps). Apart from their fundamental interest, naturally canalized PhPs may become key elements for photonic applications exploiting light–matter interactions at the nanoscale, including nanoresonators, which would greatly benefit from their strongly directional and low-loss nature.

## Methods

### Fabrication of metal antennas

Si samples were spin-coated with PMMA (950PMMA A2, purges from micro resist technology, Germany) and soft-baked at 180 °C for 15 min. The thickness of the coated PMMA layer was 90 nm, which was exposed with a Xenos pattern generator (XeDraw 2, Germany) in a Jeol 7100 F SEM. The best results were achieved at a dose of 180 μC/cm$^2$ with a development time of 60 s in a 1:3 MIBK to IPA ratio. The samples were rinsed with IPA and blow-dried with N$_2$. Finally, the samples were metalized with Cr/Au (5 nm/40 nm) in a Univex 500 (Oerlikon Leybold Vacuum GmbH, Germany) followed by a lift-off step in acetone, giving rise to the final metal antennas.

### LiV$_2$O$_5$ sample growth and preparation

LiV$_2$O$_5$ single crystals were grown by a flux method[38]. Bulk LiV$_2$O$_5$ crystals were mechanically exfoliated using a Nitto tape (Nitto Denko Co., SPV 224 P). A second exfoliation was performed from the tape to transparent polydimethylsiloxane (PDMS) in order to thin them down. The flakes were examined with an optical microscope to select homogeneous pieces with the desired thicknesses (around 100–200 nm) and large surface areas. The dry transfer technique was used to release the flakes on top of a SiO$_2$ (300 nm)/Si substrate.

### Fourier-transform infrared spectroscopy

The far-field infrared reflectance measurements of LiV$_2$O$_5$ crystals were performed using a Bruker Hyperion 2000 microscope coupled to a Bruker Vertex70v FTIR spectrometer equipped with a broadband MCT detector (400–8000 cm$^{-1}$) and a wide-range far-infrared beam-splitter (30–6000 cm$^{-1}$). Off-normal (×36 Cassegrain, 25° average incidence angle) polarized reflection spectra were obtained from the crystals. Both KRS5 and polyethylene wire grid polarizers were used to optimize the spectral throughput of the system at the relevant phonon frequencies. The spectra were collected with a 2 cm$^{-1}$ spectral resolution. An internal aperture in the microscope was adjusted to limit collection to an area on the crystal that was particularly free of defects such as cracking or varied thickness. All measurements were performed in reference to a gold film.

### Scattering-type scanning near-field optical microscopy

Near-field imaging measurements were performed employing a commercial scattering-type Scanning Near Field Optical Microscope (s-SNOM) from Neaspec GmbH, equipped with a quantum cascade laser from Daylight Solutions (890–1140 cm$^{-1}$). Metal-coated (Pt/Ir) atomic force microscopy (AFM) tips (ARROW-NCPt-50, Nanoworld) at a tapping frequency $\Omega$ ~ 280 kHz and an oscillation amplitude ~ 100 nm were used as source and probe of polaritonic excitations. Both the gold antennas and the AFM tip were illuminated with s-polarized infrared light from the quantum cascade laser. The incoming electric field was

focused at the apexes of the antennas, thus acting as two independent point dipoles. The light scattered by the tip was focused by a parabolic mirror into an infrared detector (Kolmar Technologies). Demodulation of the detected signals $n\Omega$, which can be written as the complex-valued functions $\sigma_n = s_n e^{i\phi_n}$, was performed to the 3rd harmonic ($n = 3$) of the tip frequency for background suppression. A pseudo-heterodyne interferometric method was employed to independently extract both amplitude ($s_3$) and phase ($\phi_3$) signals.

### Full-wave numerical simulations

The full-wave numerical simulations were performed using the software COMSOL Multiphysics, based on the finite boundary elements method. In the first type of simulations (Fig. 1 and Fig. 3c–e), the structure was composed of 2 semi-infinite media (superstrate and substrate) with a $LiV_2O_5$ flake in between and a vertically oriented electric dipole on top of the flake acting as a polaritonic launcher. In the second type of simulations (Fig. 4b–g), the structure was composed of a semi-infinite superstrate (air), a $LiV_2O_5$ flake with a gold antenna on top, and a semi-infinite $SiO_2$ substrate. The flake thickness has been chosen to fulfill the experimental conditions. The antenna-based simulations were based on the far-field illumination of the structure by a normally-incident plane wave polarized across the longitudinal direction of the metallic launcher. The permittivity of $SiO_2$ was taken from Ref. 39.

### Transfer−matrix numerical simulations

The transfer−matrix (TM) method based on Ref. 37 was used to obtain the polaritonic dispersion of the $LiV_2O_5$ flake along the $a$ and $b$ crystal directions. We have computed the imaginary part of the reflection coefficient (color plots in Fig. 5a, b). The poles of the reflection coefficient determine the maxima of the color plots, which correspond to the polariton dispersion. We have considered the thickness of the $LiV_2O_5$ flake to fulfill the experimental data. The superstrate and substrate employed were air and $SiO_2$, respectively.

### Dielectric function of $LiV_2O_5$

We have obtained the principal components of the $LiV_2O_5$ permittivity tensor based on the FTIR data using the Drude-Lorenz model in (1):

$$\varepsilon_i = \varepsilon_\infty^i \prod_{j=1}^{N_i} \left( \frac{\left(\omega_{LO,j}^i\right)^2 - \omega^2 - i\gamma_{LO,j}^i}{\left(\omega_{TO,j}^i\right)^2 - \omega^2 - i\gamma_{TO,j}^i} \right) \tag{1}$$

with $i = a, b, c$, $N_i$ the number of phononic oscillators along the $i$th crystal direction, $\omega_{TO,j}^i$ ($\omega_{LO,j}^i$) the transverse (longitudinal) optical phonon frequency of the $j$th oscillator along the $i$th direction, and $\gamma_{TO,j}^i$ ($\gamma_{LO,j}^i$) the damping of the transverse (longitudinal) optical phonon frequency of the $j$th oscillator along the $i$th direction. $\varepsilon_\infty^i$ is the high-frequency dielectric permittivity along the $i$th direction. The values of all the parameters were obtained by fitting the FTIR data to the Drude−Lorenz model (see Supplementary Figs. 1–3 and Supplementary Table 1).

### Analytical dispersion of polaritons under the high-momentum approximation

The analytical calculations of the IFCs and FOMs through the text have been performed by means of the high-momentum approximation of the dispersion of polaritons, given by (2)[36]:

$$q_p = \frac{\rho}{k_0 d} \left[ atan\left(\frac{\varepsilon_S \rho}{\varepsilon_c}\right) + atan\left(\frac{\varepsilon_s \rho}{\varepsilon_c}\right) + \pi l \right] \tag{2}$$

where $q_p = \frac{k_p}{k_0}$ stands for the normalized in-plane wavevector, $k_0$ and $d$ are the free-space light wavevector and flake thickness, respectively; $\varepsilon_S$ and $\varepsilon_s$ are the superstrate and substrate

permittivity, and (3)

$$\rho = \sqrt{-\frac{\varepsilon_c}{\varepsilon_a \cos^2\varphi + \varepsilon_b \sin^2\varphi}} \tag{3}$$

is a frequency and angular dependent parameter that contains the permittivity tensor components of the biaxial slab ($\varepsilon_a$ and $\varepsilon_b$ for the in-plane components and $\varepsilon_c$ for the out-of-plane component) as well as the in-plane angle $\varphi$. In all the calculations, we assume that the real and imaginary components of the IFCs are collinear. We have imposed the following condition (4):

$$Re(q_p) > 1 \tag{4}$$

to ensure that the PhPs are confined below the diffraction limit. In addition, we have set the constraint (5):

$$\left| \frac{Re(q_p)}{Im(q_p)} \right| > 1 \tag{5}$$

to consider that the PhP is propagating-like and thus feasible to measure experimentally.

## Data availability

The authors declare that the data supporting the findings of this study are available within the paper and its supplementary information. All raw data generated during the current study are available from the corresponding authors upon request.

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

## Acknowledgements

A.I.F.T.-M. and G.À.-P. acknowledge support through the Severo Ochoa program from the Government of the Principality of Asturias (nos. PA-21-PF-BP20-117 and PA-20-PF-BP19-053, respectively). J.T.-G. acknowledges support from the Swiss National Foundation (grant No. 200020_201096). M.V. acknowledges support by MCIN/AEI/10.13039/501100011033/FEDER,UE under grant PID2022-136784NB and by the Asturias FICYT under grant AYUD/2021/51185 with the support of FEDER funds. J.R.M. acknowledges support from the National Science Foundation under grant NSF-DMR-1904793. J.M.-S. acknowledges financial support from the Ramón y Cajal Program of the Government of Spain and FSE (RYC2018-026196-I), the Spanish Ministry of Science and Innovation (State Plan for Scientific and Technical Research and Innovation grant no. PID2019-110308GA-I00/AEI/10.13039/501100011033) and project PCI2022-132953 funded by MCIN/AEI/10.13039/501100011033 and the EU "NextGenerationEU/PRTR". A.Y.N. acknowledges the Spanish Ministry of Science and Innovation (grant PID2020-115221GB-C42) and the Basque Department of Education (grant PIBA-2023-1-0007). J.D.C. acknowledges support from the Office of Naval Research, Multi-University Research Initiative (MURI) program on "Twist-Optics" under grant number N00014-23-1-2567, while J.R.M. was supported under grant number N00014-22-1-2035. P.A.-G. acknowledges support from the European Research Council under Consolidator grant No. 101044461, TWISTOPTICS, and the Spanish Ministry of Science and Innovation (State Plan for Scientific and Technical Research and Innovation grant number PID2022-141304NB-I00).

## Author contributions

P.A.-G. conceived and supervised the work. A.I.F.T.-M. performed the sample fabrication, near-field experiments, and numerical simulations with the help of J.T.-G., A.T.M.-L., J.M.-S., and J.D. S.P. and M.V. performed the fabrication of Au antennas. M.I. grew the single crystals. C.L. performed the analytical study and simulations with input from G.À.-P. J.R.M. performed the far-field FTIR measurements and extracted the dielectric permittivity relation, while J.D.C. supervised the far-field experiments and analysis. P.A.-G., A.I.F.T.-M., and C.L. wrote the paper with input from G.Á.-P, A.Y.N, and J.D.C. All authors contributed to the scientific discussion and paper revisions.

## Competing interests

The authors declare no competing interests.
