## [Peer Review File · Nature Communications]

Observation of Naturally Canalized Phonon Polaritons in
 LiV_2O_5 Thin LayersEditorial Note: This manuscript has been previously reviewed at another journal that is not operating a transparent peer review scheme. This document only contains reviewer comments and rebuttal letters for versions considered at *Nature Communications*.

REVIEWER COMMENTS

Reviewer #2 (Remarks to the Author):

The authors have addressed my comments, except for one.

It relates to the determination of the dielectric constant along “c” from the two spectra presented in Supplementary. The data presented does not (yet) convincingly place a reststrahlen band between 980 and 1014 cm⁻¹. Did the authors measure multiple incident angles? In which direction is the off-normal tilt applied? Does it influence the reflectivity measured? Does the feature at ~1020 cm⁻¹ in b-polarization disappear at normal incidence? Is this the c-polarized oscillator? Since the position of this reststrahlen band is an important aspect of this work and may influence further works on this material, I believe this aspect should be firmly established.

Also, in their reply, the authors mention the FTIR reflectivity and Raman published in Physical Review B 61, 11454 (2000). However, this reference is found in the text in a sentence mentioning (more or less) that the crystal is orthorhombic. I think that more “credit” should be given to this work. This reference should appear again and a few sentences like those provided in the authors’ response should be added.

Reviewer #3 (Remarks to the Author):

In this manuscript the authors present a comprehensive study on the behavior of phonon polaritons in a novel van der Waals material LiV₂O₅. They demonstrate several unique properties, including strong field confinement, slow group velocity, and extended lifetime. Most significantly, the manuscript presents the phenomenon of absorption-facilitated canalization in PhPs. While this concept has been theoretically proposed in plasmons on two-dimensional materials and metamaterials, I believe its experimental demonstration in the context of PhPs is a novel development. Therefore, I recommend the publication of this manuscript. I would like to offer two additional comments that could further enhance the clarity of the manuscript.

First, as raised in previous review comments, concerning the issue of limited visibility of oscillations in Figure 4, I am convinced by the authors' explanation attributing it to the ultra-slow group velocity of PhPs. The new results utilizing an Au antenna with a perpendicular orientation also strengthen the claims. However, in the new experiment, that canalized PhPs should be launched on both sides of the Au-antenna. It would be beneficial if the authors plot both sides of the antenna and such a demonstration would make the experiments more compelling. Additionally, the authors should provide a discussion for their choice of rectangular-shaped antennas. It appears that employing a circular Au disk launcher that commonly used in current s-SNOM experiments, which exhibits 360-degree propagation behavior of the PhPs (like shown in the simulations of Figure 1), would more effectively illustrate the

canalization phenomenon.

Second, recent literatures have demonstrated the directional propagation of PhPs in natural crystals where the so-called “shear polaritons” are supported, see [J. Matson et al, Nature Communications, volume 14, 5240, 2023]. While this phenomenon has not been explicitly named “canalization,” it appears to align with the authors' definition of collimation of the flux of energy along a single axis. This effect also exists in natural crystals like Ga₂O₃ and CdWO₄ without the need for complex fabrication processes. Therefore, the novelty of the current work could be further justified if the authors add a comparison between the absorption-facilitated canalization and this related phenomenon to illustrate the advantages of this work

Response to Reviewer #2:

We thank the reviewer for suggesting a detailed explanation of the determination of the out-of-plane component of the permittivity tensor, as well as for helping to improve the manuscript by giving credit to the reference Physical Review B 61, 11454 (2000).

(1). It relates to the determination of the dielectric constant along “c” from the two spectra presented in Supplementary. The data presented does not (yet) convincingly place a reststrahlen band between 980 and 1014 cm^{-1} . Did the authors measure multiple incident angles? In which direction is the off-normal tilt applied? Does it influence the reflectivity measured? Does the feature at $\sim 1020 \text{ cm}^{-1}$ in b-polarization disappear at normal incidence? Is this the c-polarized oscillator? Since the position of this reststrahlen band is an important aspect of this work and may influence further works on this material, I believe this aspect should be firmly established.

We agree with the reviewer on the importance of a more complete analysis of the FTIR spectra around 1020 cm^{-1} to convincingly define their origin and thus the reststrahlen band between 980 and 1024 cm^{-1} . In the revised manuscript we include new reflectance measurements using three objectives with different nominal incidence angles: 36x objective with an incidence of 25 degrees, 15x Cassegrain objective with an incidence of 17 degrees, and grazing-incidence objective with an incidence of 55 degrees. The results, shown in the new Fig. S2, clearly reveal that the reflectance peak around 1020 cm^{-1} weakens with decreasing incidence angle, allowing us to corroborate its origin from the response of a c-polarized oscillator. Thus, despite the challenges posed by the thin nature of the sample, which makes it difficult to accurately determine the properties of the out-of-plane phonons, this observation allows us to identify the presence of the c-oriented optical phonon.

These new results have been added to Supplementary Note I as follows:

We conducted reflectance measurements (along the bc-axis) at different incident angles for comparative analysis. Typically, a 36x objective is used for FTIR spectroscopy on this type of sample due to its small size. This objective has a nominal incidence of 25 degrees, with an angular spread. The result of this objective is depicted by the black solid line in Fig. S2. In addition, results obtained with a 15x Cassegrain objective with a nominal incidence of 17 degrees (blue solid line) and a grazing-incidence objective with a nominal incidence of 55 degrees (red solid line) are also shown.

Fig. S2. Reflectance FTIR measurements were obtained using three objectives with different nominal incident angles: 36x objective with a 25-degree incidence (black solid line), 15x Cassegrain objective with a 17-degree incidence (blue solid line), and grazing-incidence objective with a 55-degree incidence (red solid line). **A**, Reflectance measured in the spectral range from 750 to 1200 cm^{-1} . **B**, Zoom-in of A from 950 to 1100 cm^{-1} .

Fig. S2 clearly reveals that the reflectance peak at around 1020 cm^{-1} weakens with decreasing incidence angle, allowing us to corroborate its origin from the response of a c-polarized oscillator. Thus, despite the challenges posed by the thin nature of the sample, which makes it difficult to accurately determine the properties of the out-of-plane phonons, this observation allows us to identify the presence of the c-oriented optical phonon. The provided phonon parameters (reference ³⁴ from the main text) yield a good fit for the dielectric function. We have refrained from including the dielectric function model spectra here, as the other two spectra were unable to fully capture the flake size (due to lower magnification objectives), making it challenging to completely remove the background.

(2). Also, in their reply, the authors mention the FTIR reflectivity and Raman published in Physical Review B 61, 11454 (2000). However, this reference is found in the text in a sentence mentioning (more or less) that the crystal is orthorhombic. I think that more “credit” should be given to this work. This reference should appear again and a few sentences like those provided in the authors’ response should be added.

As suggested by the reviewer, we have included the mentioned reference in the following sentences of the main text, denoted by the index ³⁴:

“Resulting from the intercalation of $\alpha\text{-V}_2\text{O}_5$ with the alkaline atom Li^{33,34}, [...]”.

“The lattice parameters are $a = 0.9702$ nm, $b = 0.3607$ nm, and $c = 1.0664$ nm along the [100], [001], and [010] crystal directions, respectively³⁴, [...]”.

“Previous studies provided the initial dielectric function fit values³⁴. To ensure robust agreement, these values were then tailored to align with our experimental spectra”.

Also, we cite it in the following sentence of the Supplementary Information:

“The provided phonon parameters (reference ³⁴ from the main text) yield a good fit for the dielectric function”.

Response to Reviewer #3:

We thank the reviewer for recommending the publication of our manuscript. Particularly, we would like to express our gratitude for comments such as “*the authors present a comprehensive study on the behavior of phonon polaritons in a novel van der Waals material LiV₂O₅*” and “*I believe its experimental demonstration in the context of PhPs is a novel development*”. Below, we carefully address the concerns raised, which have particularly contributed to a more extensive explanation of the launching of PhPs and the particularities of “natural” canalization.

(1). “First, as raised in previous review comments, concerning the issue of limited visibility of oscillations in Figure 4, I am convinced by the authors' explanation attributing it to the ultra-slow group velocity of PhPs. The new results utilizing an Au antenna with a perpendicular orientation also strengthen the claims. However, in the new experiment, that canalized PhPs should be launched on both sides of the Au-antenna. It would be beneficial if the authors plot both sides of the antenna and such a demonstration would make the experiments more compelling. Additionally, the authors should provide a discussion for their choice of rectangular-shaped antennas. It appears that employing a circular Au disk launcher that commonly used in current s-SNOM experiments, which exhibits 360-degree propagation behavior of the PhPs (like shown in the simulations of Figure 1), would more effectively illustrate the canalization phenomenon”.

We thank the reviewer for giving credit to our explanation regarding the low-loss nature of PhPs in LiV₂O₅. In our experiments, an elongated Au antenna is used to launch PhPs. This choice lies in the fact that, at the apexes of the antenna, a dipolar mode is generated, and thus the apexes act as point-like dipoles. This is an optimal source for launching PhPs. Several examples can be found elsewhere. Below, we show some well-known works in polaritonics that use this configuration:

- Pons-Valencia, P., Alfaro-Mozaz, F.J., Wiecha, M.M. et al. Launching of hyperbolic phonon-polaritons in h-BN slabs by resonant metal plasmonic antennas. *Nat Commun* 10, 3242 (2019).
- Zheng, Z. et al. Phonon Polaritons in Twisted Double-Layers of Hyperbolic van der Waals Crystals. *Nano Lett.* 20 (7), 5301-5308 (2020).
- Duan, J. et al. Twisted Nano-optics: Manipulating Light at the Nanoscale with Twisted Phonon Polaritonic Slabs. *Nano Lett.* 20, 5323-5329 (2020).
- Duan, J., Álvarez-Pérez, G., Lanza, C. et al. Multiple and spectrally robust photonic magic angles in reconfigurable α -MoO₃ trilayers. *Nat. Mater.* 22, 867–872 (2023).

However, due to lithographic drawbacks, one of the apexes of the antennas is not so well defined in our last experiment but has a circular-like shape and thus becomes an extended source. This fabrication imperfection (see the right image in Fig. R1 below) significantly reduces the field enhancement at this side of the antenna and thus importantly affects the dipolar mode efficiency of the apex. Based on this, we would prefer not to add a full image of the antenna in the main text, but to focus on the apex which launches PhPs more efficiently. To better visualize and compare the launching efficiency from these elongated Au nanoantennas, we have added below (Fig. R1) two near-field measurements

performed with these antennas oriented along the a-direction (left) and b-direction (right). Clearly, the launch efficiency from the lower apex of the b-direction-oriented nanoantenna is greatly reduced, which we attribute to fabrication imperfections. However, the upper apex allows the launching and visualization of canalized PhPs more clearly than when using the nanoantenna oriented along the perpendicular direction (left).

Fig. R1. Near-field images of LiV_2O_5 PhPs (s_3) launched by two different rectangular antennas at $\omega = 1010 \text{ cm}^{-1}$ (left) and $\omega = 1006 \text{ cm}^{-1}$ (right).

As previously stated, a rod-like geometry for the nanoantenna acts as a more efficient PhP launcher than other geometrical configurations, such as e.g., a circular antenna. To better appreciate this difference, we have conducted new experiments in which a circular nanoantenna was used as PhP launcher. These results are included in the revised Supplementary Information as follows:

Supplementary Note VII. s-SNOM images of LiV_2O_5 PhPs launched by circular nanoantennas

In this section, we show s-SNOM measurements of PhPs in LiV_2O_5 launched by a circular nanoantenna fabricated on top of a single LiV_2O_5 flake. We can observe a clear transition (left to right in Fig. S12) in the PhP propagation from elliptical to canalized wavefronts. Although this experiment allows us to unveil natural canalization in LiV_2O_5 , the launching efficiency of the circular antennas is clearly not as good as when using a rectangular nanoantenna (Fig. 4 of the main text), where the strong dipolar mode localizes the near fields at its apices (thus acting as point-dipole sources). In the case of the circular nanoantennas, the source is no longer point-like but extended, leading to a less efficient launching. Moreover, the surface of the circular antennas is rather rough, which may lead to a decrease in their scattering effectiveness.

Fig. S12. **A**, Near-field s-SNOM measurements of LiV_2O_5 PhPs (s_3) launched by a circular nanoantenna at $\omega = 997 \text{ cm}^{-1}$ (left), $\omega = 1001 \text{ cm}^{-1}$ (middle) and $\omega = 1006 \text{ cm}^{-1}$ (right). The diameter of the nanoantenna is $\sim 2.5 \mu\text{m}$, and its height is $\sim 40 \text{ nm}$. The thickness of the flake is $d = 213 \text{ nm}$. **B**, Same images as in **A** but with dashed gray lines added as guides to the eye.

These experimental results are also supported by numerical simulations (Fig. R2) in which canalized PhPs at $\omega = 1007 \text{ cm}^{-1}$ are excited by either a circular (left) or a rectangular nanoantenna (right):

Fig. R2. Numerical simulations of circular (left) and rectangular (right) Au nanoantennas fabricated on top of a single LiV_2O_5 flake (thickness $d = 116 \text{ nm}$).

As we observe in Fig. R2, canalized polaritons along the b -direction are launched by both nanoantenna geometries. However, the circular antenna acts as an extended source, formed by an infinite number of point-like dipoles located along its edge.

(2). “Second, recent literatures have demonstrated the directional propagation of PhPs in natural crystals where the so-called “shear polaritons” are supported, see [J. Matson et al, Nature Communications, volume 14, 5240, 2023]. While this phenomenon has not been explicitly named “canalization,” it appears to align with the authors' definition of collimation of the flux of energy along a single axis. This effect also exists in natural crystals like Ga2O3 and CdWO4 without the need for complex fabrication processes. Therefore, the novelty of the current work could be further justified if the authors add a comparison between the absorption-facilitated canalization and this related phenomenon to illustrate the advantages of this work”.

We thank the reviewer for suggesting a further discussion on the concept of canalization in comparison to recently reported shear-like polaritons.

As the reviewer correctly comments, recent works (Matson, J., Wasserroth, S., Ni, X. *et al.* Controlling the propagation asymmetry of hyperbolic shear polaritons in beta-gallium oxide. *Nature Communications* **14**, 5240 (2023), C. Passler, N. *et al.* Hyperbolic shear polaritons in low-symmetry crystals. *Nature* **602**, 595-600 (2022) or Hu, G. *et al.* Real-space nanoimaging of hyperbolic shear polaritons in a monoclinic crystal. *Nature Nanotechnology* **18**, 64-70 (2023)) have demonstrated shear- and ray-like PhP propagation that could be seen as “pseudo-canalized” (read below) PhPs. However, there are important differences with our current result that we would like to highlight:

Firstly, shear-like polaritons are measured at structures made of 2 “quasi” semi-infinite media; i.e., a superstrate (air) and a wafer (and thus very thick). In these systems, due to their size, PhPs are usually low-confined with wavevectors of the order of the free-space light. In contrast, we have used very thin layers of the orthorhombic material LiV₂O₅ (in the order of hundreds of nanometers). This will ultimately lead to very subdiffractive light, as reported in the main text, where the polariton wavelength can be confined up to $\lambda_p \sim \lambda_0/27$, as demonstrated experimentally.

More importantly, “shear” PhPs display ray-like propagation, which consists of an “infinite” number of “high-valued” wavevectors, implying a close-to-negligible PhP wavelength. Particularly, rays are confined to the interface, as recently reported in <https://doi.org/10.1021/acsp Photonics.3c01428>. In our experiments, the polaritonic modes are volume-like along the a and b directions of the crystal, in the sense that the PhP electromagnetic field is not confined to the air-LiV₂O₅ interface but is able to penetrate the substrate. Thus, when natural canalization appears along the b-direction, PhP modes remain volume-like.

Finally, the propagation of the “shear” rays is not restricted to a unique effective crystal axis, but displays propagation along two main directions, leading to a cross-like pattern in real space. Propagation could occur in a single direction, as demonstrated in these recent works, but only in one sense. They still exhibit hyperbolic propagation in the other sense, not as an intrinsic property of the material, but as an effect from the incoming illumination. In our work, the propagation of PhPs is restricted to a single crystal direction, we observe canalization independently of the illumination, as an intrinsic property. A further discussion on the combination of materials supporting naturally canalized polaritons and shear-like hyperbolic polaritons can be the subject of future works, in which the search for very anisotropic PhPs leads to fascinating phenomena. We have added the references (Matson, J., Wasserroth, S., Ni, X. *et al.* Controlling the

propagation asymmetry of hyperbolic shear polaritons in beta-gallium oxide. *Nature Communications* **14**, 5240 (2023), C. Passler, N. *et al.* Hyperbolic shear polaritons in low-symmetry crystals. *Nature* **602**, 595-600 (2022) and Hu, G. *et al.* Real-space nanoimaging of hyperbolic shear polaritons in a monoclinic crystal. *Nature Nanotechnology* **18**, 64-70 (2023)) to our main text and added the following sentence:

“Notably, this has been made possible by the emergence of novel polaritonic media, including vdW materials such as h-BN,¹⁵⁻¹⁸ α -V₂O₅¹⁹, and α -MoO₃^{20,21}, and other bulky crystals such as β -Ga₂O₃^{22,23} or CdWO₄²⁴, all of them supporting highly anisotropic PhPs^{25,26} – light coupled to lattice vibrations –.”

REVIEWERS' COMMENTS

Reviewer #2 (Remarks to the Author):

[Note from the editor: the reviewer only relayed confidential comments to the editor.]

Reviewer #3 (Remarks to the Author):

I appreciate the authors' efforts in addressing my comments raised in the previous review. They have made a persuasive case in their response and have enhanced the quality of their manuscript. Notably, the additional experimental and simulation results justify their decision to utilize rectangular-shaped antennas. The authors' discussion on the differences between absorption-facilitated canalization and hyperbolic shear polaritons is also convincing. Consequently, I am pleased to recommend the manuscript for acceptance.